# Identification and characterisation *of Mansonella perstans* in the Volta Region of Ghana

**Millicent Opoku** [1,2], **Dziedzom K. de Souza** [1] *

1 Department of Parasitology, Noguchi Memorial Institute for Medical Research, University of Ghana, Accra, Ghana, 2 Environment and Genetics Department, School of Agriculture, Biomedicine and Environment, La Trobe University, Melbourne, Australia

* ddesouza@noguchi.ug.edu.gh

## Abstract

*Mansonella spp*. have been reported to have a wide global distribution. Despite the distribution and co-occurrence with other filarial parasites like *Wuchereria bancrofti*, *Onchocerca volvulus* and *Loa loa*, it is given little attention. There are few surveillance programmes for assessing the distribution of mansonellosis, due to the associated mild to no symptoms experienced by infected people. However, addressing this infection is critical to the onchocerciasis control program as current rapid diagnostic tools targeting *O. volvulus* have the tendency to cross react with *Mansonella species*. In this study we identified and characterised *M. perstans* from five sites in two districts in the Volta Region of Ghana and compared them to samples from other regions. Night blood smears and filter blood blots were obtained from individuals as part of a study on lymphatic filariasis. The Giemsa-stained smears were screened by microscopy for the presence of filarial parasites. Genomic DNA was extracted from blood blots from 39 individuals that were positive for *M. perstans* and Nested PCR targeting the internal spacer 1 (ITS-1) was conducted. Of these, 30 were sequenced and 24 sequences were kept for further analysis. Phylogenetic analysis of 194 nucleotide positions showed no differences in the samples collected. The similarities suggests that there could be one species in this area. However, more robust studies with larger sample sizes are required to draw such conclusions. We also observed a clustering of the samples from Ghana with reference sequences from Africa and Brazil, suggesting they could be related. This study draws further attention to a neglected infection, presents the first characterisation of *M. perstans* in Ghana and calls for more population-based studies across different geographical zones to ascertain species variations and disease distribution.

## Introduction

Mansonellosis is a neglected tropical disease predominantly prevalent in the tropics and subtropics, with notable occurrences in Latin America, including Brazil and the Caribbean, as well as in sub-Saharan Africa [1]. The greatest burden of *Mansonella* infection is observed in sub-Saharan Africa, where an estimated 114 million individuals are infected, while nearly 600

**Data Availability Statement:** The alignment file used for the phylogenetic tree can be accessed via: https://github.com/Millicen/Mansonella-perstans/blob/main/Alignment_Fig2.mas.

**Funding:** The author(s) received no specific funding for this work.

**Competing interests:** The authors have declared that no competing interest exist.

million people are at risk of contracting the disease [1,2]. It is caused by the filarial nematodes, *Mansonella perstans*, *M. ozzardi* and *M. streptocerca*. Of the three species, *M. ozzardi* is the major species in Latin America and the Caribbean, while *M. perstans* and *M. streptocerca* are reported in Africa [3]. Transmission is through the bite of infective midges (*Culicoides* spp.) and *Simulium* blackflies (transmitting *M. ozzardi* in parts of Latin America) [4–7]. In Cameroon, *C. milnei* is the primary vector for *M. perstans* [8]. *C. milnei*, *C. imicola*, and *C. inornatipennis* have been described in Benin [9]. While in Ghana, *C. inornatipennis* is highly anthropophilic and could be the major vector [10]. Following the bite of an infective vector, infective larvae penetrate the bite wound and develop into adult worms that reside in serous body cavities or subcutaneous tissues [11]. Female adult worms produce thousands of unsheathed microfilariae that exhibit no periodicity (this is the diagnostic stage) [11]. Midges take up bloodmeal, ingesting microfilariae which then grow into infective larvae that can be injected into humans to continue the cycle. There are no distinct symptoms and clinical manifestations may range from headache, pruritus, eosinophilia, to pain in bursae, joint synovia, i.e. serous cavities and abdominal pain, proptosis, swelling of the eye and nodules in the conjunctiva, resembling Calabar swelling in *Loa loa* infection have been reported [12–14]. Diagnosis is by demonstration of microfilariae in peripheral blood [15]. Unlike other filarial infections such as lymphatic filariasis (LF) and onchocerciasis there is no standard treatment for *Mansonella* infection. However, the distribution of ivermectin (IVM) through mass drug administration (MDA) campaigns for lymphatic filariasis (LF) and onchocerciasis may have impacted *M. perstans* prevalence [16]. On the contrary, there have been reports of high *M. perstans* prevalence in rainforest settings in Camerron following decades of MDA suggesting that IVM is not effective [17]. However, drugs like diethylcarbamazine, mebendazole and doxycycline (which targets the bacteria endosymbiont *Wolbachia*) have been used to successfully clear the adult worms and reduce microfilariae burden in humans [18–21].

*M. perstans* has been reported along the coastal areas of equatorial Brazil and the Caribbean [1]. A highly prevalent variant of *M. perstans* variant known as "DEUX" was also identified in Gabon [22]. Further studies suggest its relatedness to strains from Brazil and further concludes that the strains from Brazil may have diverged from "DEUX" [22,23].

The first report of mansonellosis was in 1922 where, *M. streptocerca* was identified in the skin of indigenes from Gold Coast, now Ghana [24]. Subsequently, *M. perstans* was reported in the 1991 by Awadzi and colleagues [25]. However, its pathogenicity was poorly understood. Few studies have tried to ascertain the epidemiology, with a paucity of information. Other works in the 90's also reported on the presence of *M. perstans* in the Volta Region, but, the prevalence was not known [25]. Following on, in 2014, a study reported a high burden of *M. perstans* in individuals coinfected with Buruli Ulcer that were sampled from 2010–2012 in the middle belt of Ghana [26]. This report showed a 23% *M. perstans* in Buruli Ulcer patients compared to 13% in control participants. In 2017, Debrah and colleagues reported an average of 32% and as high as 75% infection prevalence in communities in the middle belt of Ghana [10]. Whereas gender may influence the risk of infection in some settings, although more studies are needed to substantiate these differences, infections appear to increase with age [10,27–29]. Furthermore, the environment plays a key role in vector-borne disease transmission dynamics and *M. perstans* is no exception. *M. perstans* occurs in rainforest areas with dense vegetation in Africa, where tree holes and vegetation cover support the breeding of vectors [10,17,30–34]. High prevalence of *M. perstans* infections have been linked to high vegetation, swampy areas and where livestock droppings provided suitable breeding conditions for the midges [10].

Despite its wide distribution, mansonellosis tends to be neglected due to its typically mild and asymptomatic nature [35]. This lack of attention may be attributed to the limited surveillance, monitoring activities, and inadequate funding for assessing the burden and

implementing control measures [1,6]. However, it's noted that mansonellosis could impact immune regulatory mechanisms, potentially leading to immune suppression and risk of susceptibility and severe outcome of co-infections with other diseases like malaria, HIV, and tuberculosis (TB) as well as vaccine efficacy [1,6,36]. Studies in pregnant women with malaria and or HIV showed increased risk of anaemia in the presence of *M. perstans* infections [32,37,38]. These suggests a complex public health scenario that requires urgent attention.

Despite the growing evidence of *M. perstans* presence and possible contribution to filarial morbidities, it still remains understudied [35,39]. Additionally, evidence presented by some studies, indicating that 10% of all infections may lead to disease morbidity [35], calls for further studies in understanding the health impact of this widely distributed and prevalent filarial parasite [40]. Furthermore, the co-occurrence with other filarial parasites poses a great challenge to current diagnostic kits and present a need for the development of new diagnostics. For instance, cross-reactivity with serology-based rapid diagnostic test (RDT) for LF often leads to over estimations of LF prevalence, posing challenges to evaluating intervention efforts [18,41–43].This therefore underscores the importance of increased awareness, surveillance, and funding to better understand and address mansonellosis and its potential impact on public health, especially in regions where it coexists with other filarial parasites and other infectious diseases. This study aimed to characterise *Mansonella spp*. identified during a study assessing the prevalence of lymphatic filariasis in some districts in the Volta and Eastern Regions of Ghana.

## Materials and methods

### Ethics statement

The study was conducted in agreement with the International Council for Harmonization of Technical Requirements for Pharmaceuticals for Human Use (ICH) guidelines on Good Clinical Practice (GCP). Approval for the study was obtained from the Institutional Review Board of the Noguchi Memorial Institute for Medical Research (CPN 061/16-17) and the Ethics Review Committee of the Ghana Health Service (GHS-ERC 06/08/16). Written consent was obtained from all study participants. For children, written parental consent was obtained, and written assent from children 12 to 17 years old. Participation in this study was voluntary and study objectives, potential risks, benefits, and rights to discontinue were explained to the participants. Data was anonymized prior to analysis.

### Study sites

The study was conducted in 2018 in the East Akyim district (Eastern Region), Adaklu district (Volta Region) and Hohoe district (Volta Region) of Ghana. Communities with *M. perstans* are shown in Fig 1. In each district participants were recruited from 15 communities based on population proportional to estimated size. These districts experience two raining seasons, with an average rainfall of 513.9 mm to 1099.88 mm. Temperature ranges from 12°C to 32°C. The areas have varying vegetation cover ranging from Forest to savannah grassland, highland areas and in the forest zone which receive the highest rainfall and the dry Sahel savannah zone in the north [44]. The maps (in Fig 1.) were generated using open source QGIS Desktop 3.32.2. Shapefiles used are available at https://github.com/Millicen/Mansonella_perstans/blob/main/GHA_adm.zip and publicly available for download at https://www.diva-gis.org/gdata.

### Sample collection

Nighttime blood samples were collected between 21:00 and 2:00 from 50–100 randomly selected participants (aged 5 years and above) from each community. 60µl of the blood was

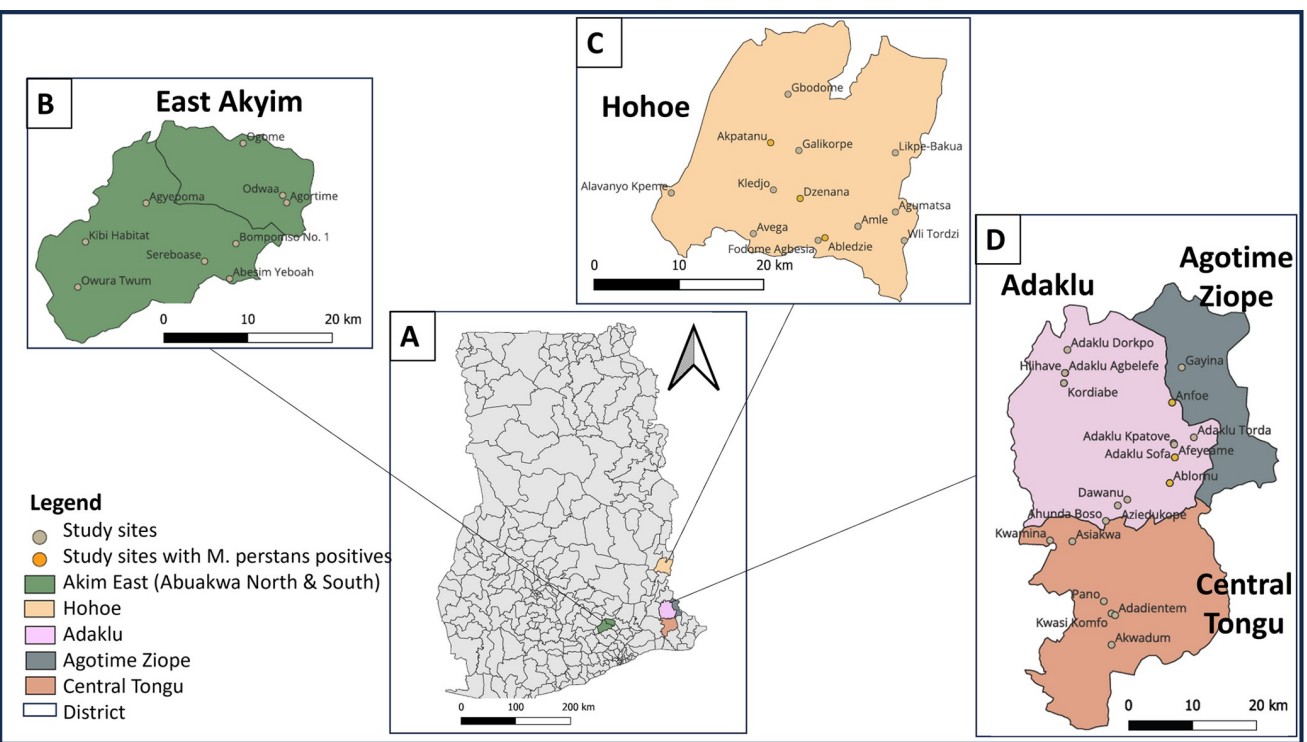

**Fig 1. Map of Ghana showing the districts and communities with *M. perstans*. A,** showing the map of Ghana with the districts where samples were collected. **B,** shows the East Akyim district (comprising the Abuakwa North and South Districts (Eastern Region)) in green with communities sampled in pale orange circles. **C,** shows the Hohoe district (Volta Region) in orange. **D,** shows three districts (Adaklu, Agotime Ziope and Central Tongu Districts in purple, grey and brown respectively) also in the Volta Region. Communities sampled are in circles with yellow and grey indicating sites with *M. perstans* positives and negatives respectively.

used for the preparation of blood films for the identification of *W. bancrofti* microfilariae (mf) and 60 μl to prepare six dry blood spots (DBS), on the six discs of the Tropbio filter paper (Tropbio Pty Ltd, QLD, Australia). The blood films were stained with Giemsa and observed under a compound microscope.

## Sample processing and analysis

Following the identification of *M. perstans* in the blood films, 39 DBS were retrieved from individuals that tested positive for *M. perstans*. DNA was extracted from 3 discs of the DBS each per participant. The extractions were done using, the Quick-DNA™ MiniPrep (Zymo Research, USA) according to the manufacturer's instructions with 75 μL elution volume. Briefly, 400 μl of Genomic Lysis Buffer containing beta-mercaptoethanol in 0.5%(v/v) was added to the DBS followed by vortexing for 4–6 seconds, and then incubated at room temperature for 10 minutes. The liquid containing the DNA was transferred to a Zymo-Spin™ IICR Column placed in a Collection Tube (ensure that no particles of the filter paper is pipetted with the sample). This was followed by centrifugation at 10,000 x g for one minute. The Collection Tube with the flow through was discarded. The Zymo-Spin™ IICR Column was transferred to a new Collection Tube. 200 μl of DNA Pre-Wash Buffer was added to the spin column and then centrifuged at 10,000 x g for one minute. 500 μl of g-DNA Wash Buffer was added to the spin column and then centrifuged at 10,000 x g for one minute. Finally, the spin column was transferred to a clean 1.5 mL microcentrifuge tube and 75 μl DNA Elution Buffer was added to the base of the spin column. This was incubated at room temperature for 5

minutes and then centrifuged at 10,000 x g for one minute to elute the DNA. The eluted DNA was stored at -20°C for future use. The sample were then screened for Filarial parasites using Nested PCR [45]).

**Detection of filarial parasites by nested PCR.** The Nested PCRs were carried out using four primers described in Tang and colleagues [45]. The primers included a universal primer (UN1-1R) that hybridise to the 5.8S ribosomal gene of all filarial species and other vertebrates (including mammals); the other three primers are universal to all filarial species of which two are forward primers (FIL-1F; used in the Nest I reaction, and FIL-2R; used in Nest II reaction), and one reverse primer (FIL-2R; used in Nest II). These primers amplify the ITS-1 region. The nest I reaction, included the FIL-1F (5`*GTGCTGTAACCATTACCGAAAGG*-3`) and UNI-1R (*5`- CGCAGCTAGCTGCGTTCTTCATCG*-3`) in a total reaction of 15 μL, with 300 nM primers, 300 nM dNTPmix, 1X Go-Taq PCR Buffer, 0.033 units of Go-Taq polymerase and 3 μL of the DNA. The PCRs were run at 94°C for 3 min, 35 cycles of 94°C for 30 sec, 60°C for 30 sec and 72°C for 30 sec, and 72°C for 7 min. The nest I was followed with nest II usingprimers, FIL-2F (5'-GGTGAACCTGCGGAAGGATC-3') and FIL-2R (5'-TGCTTATTAAGTCTACTTAA-3') in a 15 μL reaction, containing, 300 nM primers, 300 nM dNTPmix, 1X Go-Taq PCR Buffer, 0.033 units of Go-Taq polymerase and 2 μL of PCR product from the nest I reaction. The PCRs were run at 94°C for 3 min, 35 cycles of 94°C for 30 sec, 50°C for 30 sec and 72°C for 30 sec, and 72°C for 7 min.

The expected band sizes for nest I reaction, depending on the species are *Onchocerca volvulus*: 771 bp, *Mansonella perstans*: 739 bp, *Mansonella ozzardi*: 734 bp, *Wuchereria bancrofti*: 723 *bp*, *Loa loa*: 712 bp). However, these bands may not be visible if DNA concentrations are low. The expected band sizes depending on the filarial species for the nest II reaction are *O. volvulus*: 344 bp, *M. perstans*: 312 bp, *M. ozzardi*: 305 bp, *W. bancrofti*: 301 bp and *Loa loa*: 286 bp. The step-by-step protocol for the DNA extraction and Nested PCR have been deposited in protocols.io repository, http://dx.doi.org/10.17504/protocols.io.x54v928b4l3e/v1.

**Gel electrophoresis and purification.** The Nest I and II PCR products were run on a 2% agarose gel for 40 min and visualised under Dual LED Blue/White Light Transilluminator (BluPADTM, Bio-Helix co., ltd). The bands from the Nest II were excised for further purification using GF-1 Gel DNA Recovery Kit (GeneOn, Germany). The excised gel was weighed. Buffer GB was added to the gel in a ratio of 1:1v/v and incubated at 50°C until the gel had melted. The mixture was then transferred to the column and centrifuged at 10,000 x g for one minute. The column was washed with 750 μL Wash Buffer. The flow through was discarded and the column was spun at 10,000 x g to dry. 30 μL of Elution Buffer was added and incubated at room temperature for 5 minutes. The column was centrifuged at 10,000 x g for one minute to elute the DNA. The recovered DNA (5μL) were run on a 2% agarose gel to confirm if DNA has been successfully extracted. The step-by-step protocol for the gel purification and electrophoresis have been deposited in protocols.io repository, http://dx.doi.org/10.17504/protocols.io.x54v928b4l3e/v1. The DNA was stored at -20°C until they were shipped for sequencing. In total, 30 DNA samples that gave a 312 bp band (Figs 2 and S2 Table) following the Nest II reaction (including a no template control) were sent to Inqaba Biotechnologies, South Africa for Sanger sequencing.

**Sequence analysis.** Sequencing was done in 2021, with two reads obtained per sample. Raw reads were imported into Geneious Prime and poor reads were trimmed with an error probability limit of 0.05 (where more than 5% chance of error per base called are trimmed). The trimmed reads have been deposited in GenBank repository with Accession No. OR488627 to 51. The De Novo Assemble function was then used to generate consensus sequence for each pair rooted with DQ995497 (*Loa loa*). The trimmed consensus sequences were exported to MEGAX [46] to infer ancestry of the sequences. Briefly, the sequences were aligned with

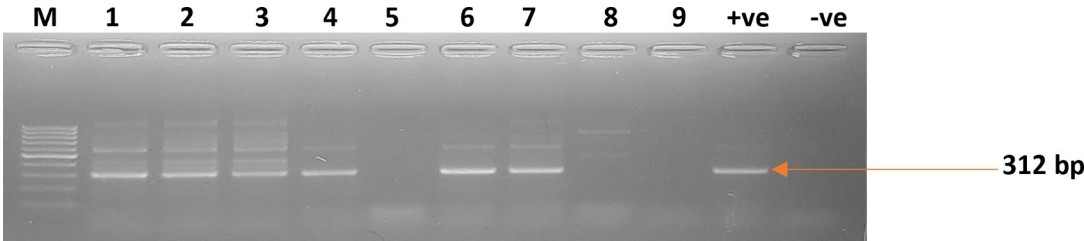

**Fig 2. Gel electrophoresis photograph showing Nest II PCR products.** Where, lanes 1 to 9 = samples; +ve = positive control; -ve = Negative control. The 312bp size shown in the sample lanes is indicative of the presence of *M. perstans* in DNA in the blood.

MUSCLE and gap open set at -400 and maximum iteration set at 16. Model selection for Maximum Likelihood and the best model estimate of the relation selected was T92+G (based on least Bayesian information criteria (BIC) and corrected Akaike information Criteria (AICc) 1546.049 and 1038.347 respectively (see S3 Table). The rate variation among sites was modelled with a gamma distribution (shape parameter = 0.58). The evolutionary history was inferred by using the Maximum Likelihood method and Tamura 3-parameter model with 500 replicates. A discrete Gamma distribution was used to model evolutionary rate differences among sites (4 categories (+G, parameter = 0.1913)). An analysis of nucleotide variance was generated using popart v1.7 software [47].

## Results

*M. perstans* was only identified in 6 communities in Hohoe (Abledzie, Akpatanu and Dzenana) and Adaklu (Ablornu, Afeyeame and Anfoe) districts (both in the Volta Region) see Fig 1, in yellow circles. The number of samples analysed from the communities are shown in S1 Table. The examination of blood slides revealed *M. perstans* infection in 33/1317 (2.51%) and 22/1355 (1.62%) individuals, respectively in these districts. The mf counts ranged from 16.7–12,350.0 mf/ml with a geometric mean of 188.6 mf/ml (S2 Table). None of the people screened in the East Akyim Districts were positive for *Mansonella* parasites (0/1074).

Due to inadequate resources and financial constraints, 39 out of the 55 microscopy positives were screened with the PCR assay. Out of the 39 samples that were processed, 29 were positive for *M. perstans*. No bands were observed for the nest I reactions. Fig 2. shows the band sizes of 312bp for nest II PCR, confirming the presence of *M. perstans* ITS-1 gene. No other species were detected.

Overall, 29 samples were sequenced using Sanger sequencing out of which 24 sequences (submitted to GenBank with Accession numbers: OR488627—OR488651) were kept for further analysis.

### Phylogenetics

An alignment using MUSCLE in MEGAX was performed with 14 reference sequences (LT623911.1, MZ285895, KR080185.1, EU272180 (*M. ozzardi*), DQ995497 (*Loa loa*), EU272181.1, EU272177.1, KR080189.1, KR080188.1, DQ995498.1, KJ631373.1, KR080187, EU272183.1 EU272182.1) are shown in Fig 3. Following the alignment, the ends were trimmed, and 194 nucleotide sites were kept for further analysis. The alignment file can be accessed via: https://github.com/Millicen/Mansonella_perstans/blob/main/Alignment.fas. Phylogenetic analysis of the sequences revealed similarity between the Ghana samples and *M. perstans* reference sequences as shown on the tree with 84% and 90% support for the branch

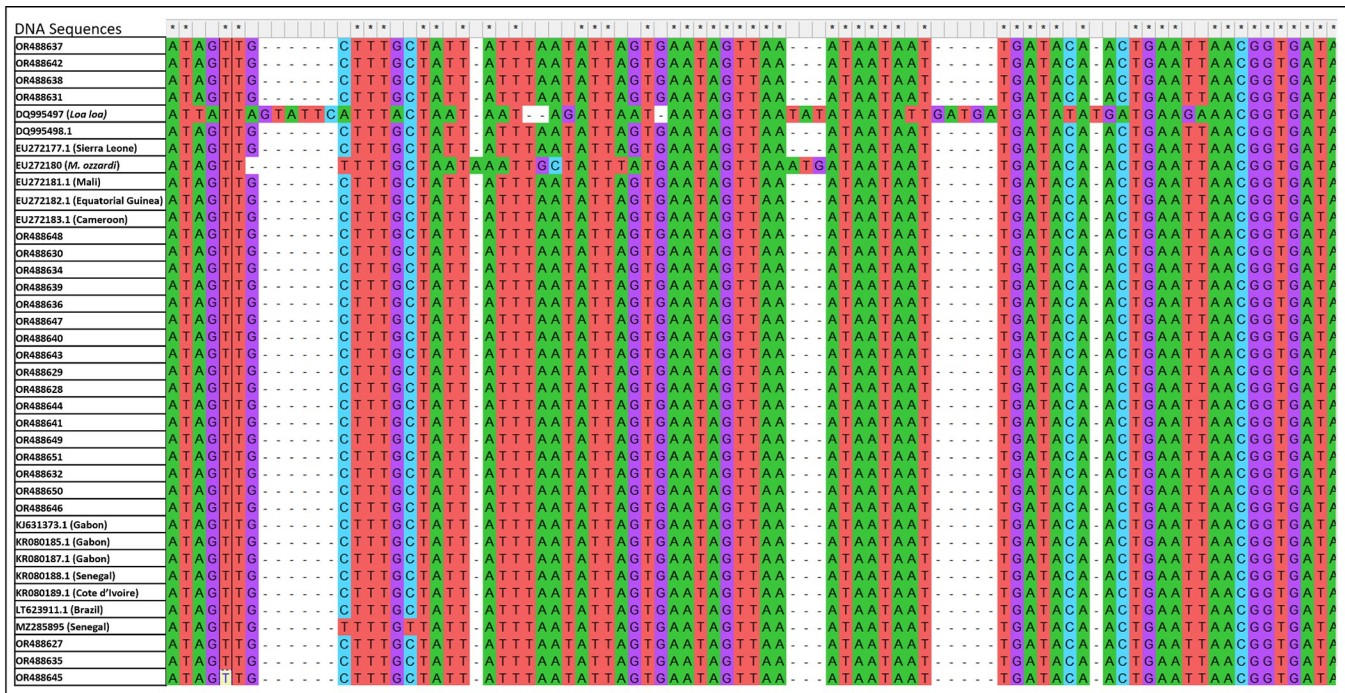

**Fig 3. Muscle Sequence alignment in MEGA X.** The alignment was generated with 24 of the newly sequenced samples and 14 reference sequences from Africa and Brazil. A total of 194 nucleotide sites were retained in the alignment.

lengths (Fig 4). KR080185.1 and MZ285895 which are reference sequences from Gabon and Senegal respectively show differences at some nucleotide positions. We did not see any population structuring in the samples obtained from the different locations in Ghana. Muscle Sequence alignment was generated in MEGA X with 24 Ghana samples and 14 reference sequences from Africa and Brazil.

## Haplotype network

An analysis of molecular variance was computed using 24 of the sequenced samples (OR488633 was excluded due to the short length) references, and five *M. perstans* of which two were from Gabon (KR080185.1 and KJ631373.1), and one each from the Amazona State in Brazil (LT623911.1), Senegal (MZ285895) and Sierra Leone (EU272177.1). A *Loa loa* sequence was included as an outgroup (DQ995497) to allow for comparison of the ingroups and due to lack of high-quality *M. perstans* reference sequences see Fig 5. The nucleotide diversity index was π = 0.00898. A significantly high variation (98.24%) among populations, but low variation (1.76%) within populations, were observed with a total variance of 11.89% (fixation index (Phi_ST) = 0.982, p = 0.051). The observed variation was based on 22 bases where, one of the references from Gabon (KJ631373.1) and that of Senegal differed at one and two nucleotide bases respectively, whereas the outgroup had 19 base differences in respect to the Ghana samples. Also, little to no variations were observed between the KJ631373.1 (Gabon), LT623911.1(Brazil) and EU272177.1 (Sierra Leone) and the Ghana samples. No variations were seen in the Ghana samples (p < 0.001).

## Discussion

In this study, sequences from individuals with *M. perstans* were analysed. We present the first characterisation of *M. perstans* internal transcribed spacer 1 (ITS-1) gene in Ghana. The

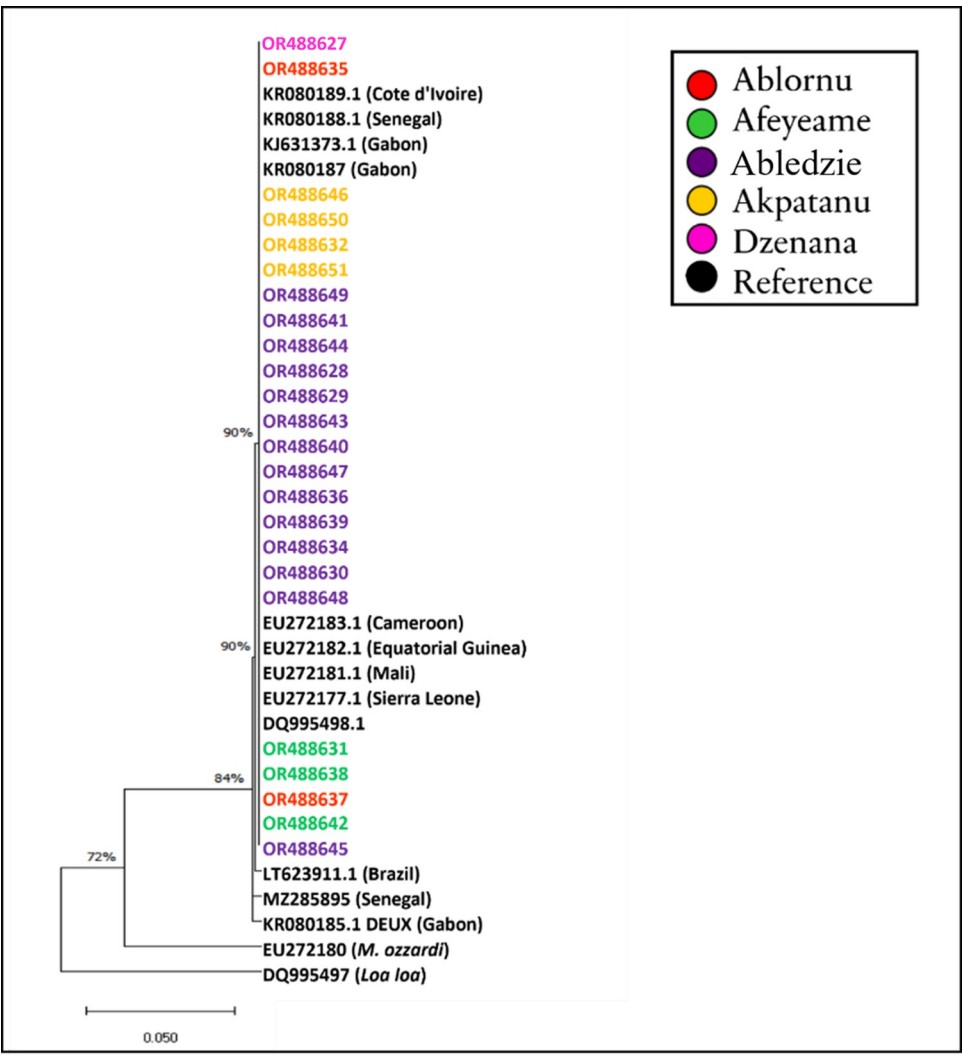

**Fig 4. A maximum likelihood tree generated in MEGAX.** Varius reference *M. perstans* sequences from Africa (including, Gabon, Senegal, Mali, Sierra Leone, Cote d'Ivoire, Equatorial Guinea and Cameroon) and Brazil, *M. ozzardi* were included with DQ995497 (*Loa loa*) as outgroup. The Ghana samples (OR488627 to 51) are coloured by location. The tree excluded OR488633 due to it short length.

Ghana samples showed no variations per the phylogenetic analysis (Fig 4) suggesting no geographical isolation in the population of *M. perstans* and hence maybe considered as one transmission zone with similar ecological conditions. However, given the small sample sizes with low numbers of representation per site, we are unable to draw firm conclusions on the populations. Nonetheless, our observation of the lack of variations in the sequences from this study in Ghana suggests that there could be only one species of *M. perstans* responsible for infections in the Volta Region. However, this needs to be substantiated with extensive sampling to include other study sites to understand *M. perstans* prevalence in the Volta Region.

Many sources of *M. perstans* infections in the world have been linked to Africa. In Italy and Spain, *M. perstans* were identified in immigrants from sub-Saharan Africa [48]. Infections in the Caribbean and the Americas, were linked to the transatlantic slave trade [23]. The clustering of the Ghana samples and other sequences from Gabon and Sierra Leone with the *M.*

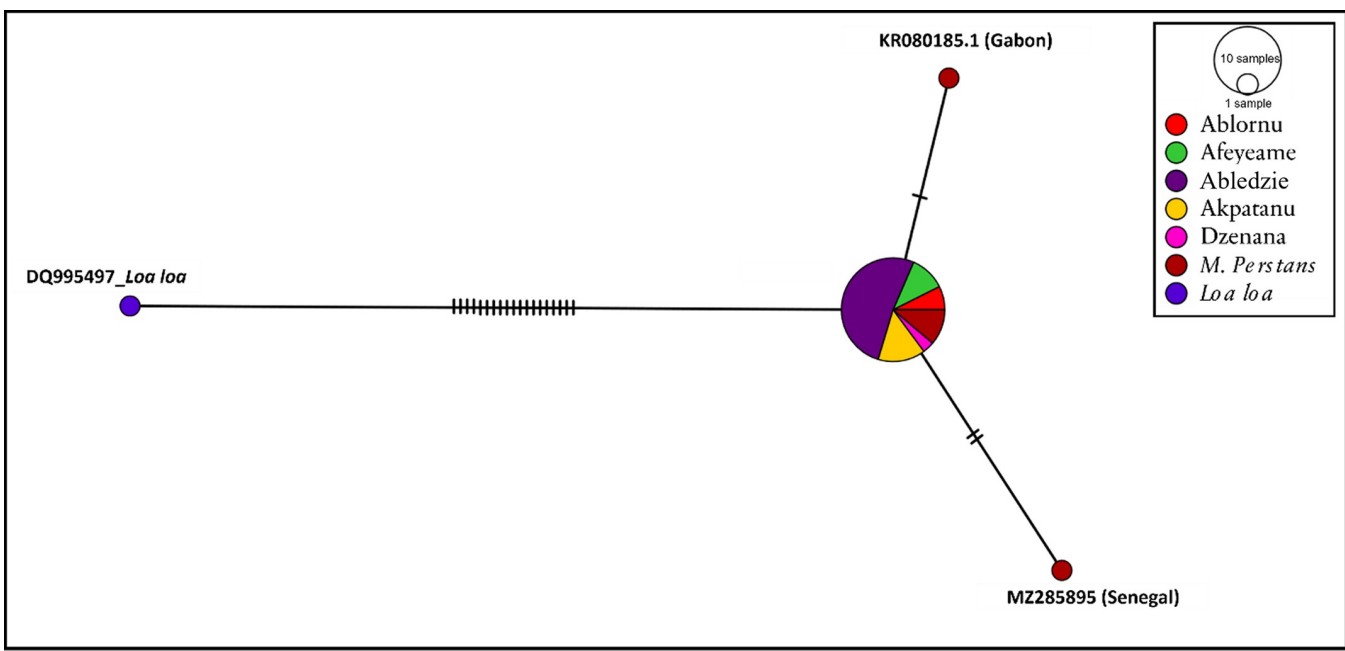

**Fig 5. Haplotype network showing nucleotide diversity.** Samples are coloured by location. Red and blue nodes indicate reference sequences of *M. perstans* and *Loa loa* respectively obtained from the NCBI.

*perstans* ITS-1 (LT623911.1) from Brazil Amazona State (Fig 4) and other countries in Africa provides evidence of association. Findings of Tavares da Silva and colleagues [23] also showed that a recently published genome of *M. perstans* from Brazil shared similarity with those from Africa. Their work supported the origin of *M. perstans* from Africa into Latin America [23] following divergence from *M. perstans* "DEUX" clade from Gabon. The genetic diversity observed in Fig 5 specifically at 1 and 2 nucleotide base differences between the Gabon ("DEUX") and Senegal isolates respectively, and those from Ghana samples suggests potential variations in the genetic makeup of *M. perstans* populations in different geographic regions. The observed genetic differences as indicated by variations in nucleotide bases, raise the intriguing possibility that these discrepancies may delineate distinct species of *M. perstans* (Fig 5). Additionally, recent publication of the *M. perstans* genome [49] will enhance further studies and understanding of this very neglected parasite. Furthermore, the availability of these genomes now provides an avenue for comparative genomics studies of the different *M. perstans* clades and their pathogenesis.

Low infection rates were observed in the study participants. Perhaps if all the blood samples had been screened by a more sensitive assay, there could have been more positives as standard microscopy tests are usually less sensitive especially as infection wanes [50–52]. While the identification of *M. perstans* following microscopy was based on the ITS-1 gene, a newer and more sensitive method based on the *M. perstans* Repeat 1 (*Mp*R1) has been developed [53]. Nonetheless, Eberhardt and colleagues also reported low prevalence of *M. perstans* in the serum of HIV patients in Ghana [54]. The low infection levels observed in the this and other studies could also be due to past ivermectin distribution for lymphatic filariasis and onchocerciasis control in Ghana. Although infections may be low now, the interruption of LF and onchocerciasis in these areas and subsequent stopping of MDA, may result in mansonellosis resurgence in future.

Furthermore, the environment plays a crucial role in vector borne disease dynamics. Whereas areas with high vegetation index with dense forests provide suitable breeding habitats for vectors consequently leading to high vector densities and high transmission, areas with low vegetation cover or grasslands tend to have fewer suitable habitats for vector breeding. Studies in parts of Central and East Africa have reported on the association of high *M. perstans* prevalence with high vector density and distribution in areas with high vegetation index [31,32,34]. It is however unclear whether the low infection prevalence reported in this study, compared to high prevalence in the middle belt of Ghana is species specific or due to environmental variations that will allow high vector breeding potential and transmission. The Volta Region is characterised by grassland, dry Savannah, highlands and few forests [44] compared to the middle belt that has dense forests with swamps which support high vector density [10]. This therefore provides an opportunity for further investigation in the *M. perstans* in other parts of Ghana.

The Volta Region bears a significant malaria burden, as highlighted by previous studies [55,56]. The identification of *M. perstans* in the lower half of Ghana, particularly in the Volta Region, may exacerbate susceptibility to and the severity of malaria infections. Moreover, the documented prevalence of coinfections and *M. perstans*' influence on immune modulation and disease progression across Africa [6,30,36–38] emphasize the need to address mansonellosis within a broader public health initiative. The coexistence of *M. perstans* and high malaria prevalence in the Volta Region underscores the necessity for comprehensive strategies and interventions. Integrating mansonellosis surveillance with existing programs targeting lymphatic filariasis (LF), onchocerciasis and malaria in Ghana presents a practical approach. This integration not only streamlines efforts but also offers insights into the interplay between various parasitic infections and their collective impact on public health. Such initiatives would enable the development of tailored interventions to tackle the challenges posed by multiple parasitic infections in affected regions of Ghana. Ultimately, this report emphasizes the importance of a coordinated, multidisciplinary approach to address the intricate landscape of *M. perstans* infections and their implications for public health in the region.

## Conclusion

Despite the potential impact of *M. perstans* on diseases of public health importance, it receives little attention. Consequently, there is no standard treatment and surveillance to track the extent of endemicity. Although symptoms are not clearly defined it is believed to impact host immune reactions and may play a crucial role in disease pathogenesis especially where it is co-endemic with other filarial parasites. In this study, we present the first characterisation of *M. perstans* in Ghana. Furthermore, the phylogenetics analysis reveals clustering of isolates from Ghana with that of Brazil. We did not observe any differences in the sequences obtained from the study sites which could probably be due to small sample size. We therefore advocate for population-based studies with a bigger target (or whole genome sequence data) that will include more samples from different locations across different geographical settings. Additionally, the coexistence of *M. perstans* with other blood filarial worms presents a unique opportunity for screening one sample from a single individual allowing for parallel monitoring for assessing the public health impact of mansonellosis in populations.

## Supporting information

**S1 Table. Communities from Adaklu and Hohoe districts.** The number of individuals positive for *M. perstans* are shown.
(PDF)

**S2 Table. Samples that were positive for *M. perstans* following microscopy.** Those that were submitted for Sanger sequencing are indicated with *.
(PDF)

**S3 Table. Model estimation and selection inferring relationships.** T92+G model was selected to build the tree based on low BIC and AICc values.
(PDF)

**S1 Raw Image.**
(JPG)

## Acknowledgments

Dr. Jewelna Akorli provided access to Geneious Software for data analysis. Dr. Shannon Hedtke provided guidance on data analysis.

## Author Contributions

**Conceptualization:** Dziedzom K. de Souza.

**Data curation:** Millicent Opoku.

**Formal analysis:** Millicent Opoku.

**Funding acquisition:** Dziedzom K. de Souza.

**Methodology:** Millicent Opoku.

**Supervision:** Dziedzom K. de Souza.

**Writing – original draft:** Millicent Opoku.

**Writing – review & editing:** Dziedzom K. de Souza.

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
