## [Decision Letter · Decision Letter 0]

18 Dec 2023

PONE-D-23-37316Identification and characterisation of Mansonella perstans in the Volta Region of GhanaPLOS ONE

Dear Dr. Opoku,

Thank you for submitting your manuscript to PLOS ONE. After careful consideration, we feel that it has merit but does not fully meet PLOS ONE’s publication criteria as it currently stands. Therefore, we invite you to submit a revised version of the manuscript that addresses the points raised during the review process.

We look forward to receiving your revised manuscript.

Kind regards,

James Lee Crainey, Ph.D.

Academic Editor

PLOS ONE

4. We suggest you thoroughly copyedit your manuscript for language usage, spelling, and grammar. If you do not know anyone who can help you do this, you may wish to consider employing a professional scientific editing service. 

A clean copy of the edited manuscript (uploaded as the new *manuscript* file)”.

5. We noted in your submission details that a portion of your manuscript may have been presented or published elsewhere. Please clarify whether this [conference proceeding or publication] was peer-reviewed and formally published. If this work was previously peer-reviewed and published, in the cover letter please provide the reason that this work does not constitute dual publication and should be included in the current manuscript.

7. We note that Figure 1 in your submission contain [map/satellite] images which may be copyrighted. All PLOS content is published under the Creative Commons Attribution License (CC BY 4.0), which means that the manuscript, images, and Supporting Information files will be freely available online, and any third party is permitted to access, download, copy, distribute, and use these materials in any way, even commercially, with proper attribution. For these reasons, we cannot publish previously copyrighted maps or satellite images created using proprietary data, such as Google software (Google Maps, Street View, and Earth). For more information, see our copyright guidelines: http://journals.plos.org/plosone/s/licenses-and-copyright.

8. PLOS ONE now requires that authors provide the original uncropped and unadjusted images underlying all blot or gel results reported in a submission’s figures or Supporting Information files. This policy and the journal’s other requirements for blot/gel reporting and figure preparation are described in detail at https://journals.plos.org/plosone/s/figures#loc-blot-and-gel-reporting-requirements and https://journals.plos.org/plosone/s/figures#loc-preparing-figures-from-image-files. When you submit your revised manuscript, please ensure that your figures adhere fully to these guidelines and provide the original underlying images for all blot or gel data reported in your submission. See the following link for instructions on providing the original image data: https://journals.plos.org/plosone/s/figures#loc-original-images-for-blots-and-gels. 

Reviewers' comments:

Reviewer's Responses to Questions

**Comments to the Author**

1. Is the manuscript technically sound, and do the data support the conclusions?

Reviewer #1: Yes

Reviewer #2: Partly

2. Has the statistical analysis been performed appropriately and rigorously? 

Reviewer #1: Yes

Reviewer #2: N/A

3. Have the authors made all data underlying the findings in their manuscript fully available?

Reviewer #1: Yes

Reviewer #2: No

4. Is the manuscript presented in an intelligible fashion and written in standard English?

Reviewer #1: Yes

Reviewer #2: Yes

5. Review Comments to the Author

Reviewer #1: Dear Authors,

Thank you for the opportunity to review your paper titled "Identification and characterisation of 'Mansonella perstans' in the Volta Region of Ghana." This study addresses a critical and often overlooked aspect of parasitology, particularly the prevalence and characteristics of 'Mansonella perstans', a notable filarial parasite, especially in sub-Saharan Africa.

Introduction:

1. Your introduction comprehensively covers the species, distribution, transmission, clinical manifestations, and the main diagnostic methods of 'Mansonella perstans'. However, it could benefit from a clearer exposition on the current understanding (or lack thereof) of effective treatments for 'M. perstans', drawing a comparison with treatments available for onchocerciasis or lymphatic filariasis (LF) where at least one effective treatment is avaliable.

2. The third paragraph, while informative, could be restructured for better flow and clarity.

3. Is there any historical data on 'Mansonella' in Ghana before? Understanding if there was a difference in the epidemiology of the parasite before and after (now) interventions for onchocerciasis and LF (as well as population growth and associated consequences) could be insightful

Methodology:

4. (2.1 Study Site) Figure A could be more well defined, although it is readable. I could only find the Hohoe and Adaklu districts, but not East Akyium district, understandably because the latter was negative for M. perstans; however, Figure 1 should not mention that M. perstans is only present in the two former districts, as it is part of the results, not the methodology.

5. (2.3 Sample Processing and Analysis) Please provide clarity on which filarial parasites were screened using Nested PCR (O. volvulus, M. perstans and M. perstans?) and the CO1 gene. Also, are these results presented, even if negative, in the result section?

6. (2.6 Sequence Analysis) The rationale for using the 'Loa Loa' sequence instead of a 'Mansonella perstans' sequence would benefit from explanation. If the choice was due to the lack of high-quality reference sequences from 'M. perstans', this should be explicitly stated.

Results:

7. The range of microfilariae counts is indeed remarkable and adds significant value to the study. It would be beneficial to include the number of individuals screened for 'M. perstans' in East Akyim District, even if the results were negative.

8. Additionally, the placement of each figure legend immediately following its corresponding description would aid in readability and comprehension instead of all at the end of the Results section.

Discussion:

9. The first paragraph of the Discussion section might be more appropriate in the Introduction, considering its content.

10. The phylogenetic analysis suggesting no variation between the non-contiguous Hohoe and Adalku districts raises intriguing questions about the potential prevalence of 'M. perstans' in surrounding districts. This could be an interesting avenue for future research to understand the prevalence of the parasite in the Volta region. Moreover, the discussion could be expanded to explore the possibilities of integrating 'M. perstans' surveillance with existing lymphatic filariasis and onchocerciasis programmes in Ghana. It would also be beneficial to discuss the broader public health implications of M. perstans, especially in relation to co-infections with other filarial diseases.

Overall, the paper makes a valuable contribution to the field. The English could be polished to enhance the paper's clarity and readability. I look forward to seeing the revised version of this important work.

Best regards,

Reviewer #2: Opoku and Souza analysed M. perstans prevalence in the Volta region of Ghana and revealed low (up to 2.51%) infection rates. Moreover, they sequenced the samples and showed that the samples cluster with the reference sequences from other African countries and Brazil. It is very important to draw attention to this neglected filarial infection but the presented study lacks novelty and intense discussion (e.g., genome analysis). Moreover, the introduction and discussion lacks recent publication and is sometimes misleading. In my opinion, the manuscript need to be carefully re-written and updated before

Major comments:

1) Line 80-82: The authors highlight that pathogenicity is not known/was so far not assessed, but the presented data sets do also not tackle this issue. Similar to Debrah et al. the presented manuscript shows the prevalence of M. perstans in a specific region of Ghana. In summary, the infection rates are very low compared to the middle belt of Ghana. The authors need to discuss that fact also in regards to the characteristics of the environment that are needed for M. perstans. For example Wanji and colleagues from Cameroon presented over the last years several publications about that.

2) A long this line, the introduction need to be updated, such as Mansonella spp. DEUX is not mentioned (line 49-50), the transmitting vector for M. perstans (C. milnei) have been deciphered in Cameroon (line 50-51), and it should be mentioned that M. perstans has no distinct clinical symptoms and is mostly asymptomatic (line 55-57).

3) Line 64-67: Of course, mansonellosis is still neglected but the are several studies targeting Mansonella and it cannot be stated that it is only assessed during lymphatic filariasis or onchocerciasis surveillance programmes. Again, Wanji and colleagues specifically target Mansonella epidemiology, immunology and treatment efficacy (showing that ivermectin is not effective - should be included in line 68-70).

4) In my opinion, it is important to explain the applied methods more in detail. For example, the used PCR need to be shortly explained instead of only revising to a publication. What can be differentiated with this PCR (specificity/what filarial spp can be discriminated?). A long this line, Fig S1 shows a gel blot but there is no further explanation, such as what are the different bands in the sample lines.

5) In regards to 4), supplement figures/tables need to be embedded, explained and in order within the text.

6) How were the 30 samples that were sequences selected from the 39 positive individuals.

7) line 229-230: There are several studies showing the impact of M. perstans on concomitant infections (Muhangi et al. 2007; Hillier et al. 2008; Stensgaard et al. 2016), which can be explained by the strong immune modulation of M. perstans (Ritter et al. 2018).

8) Recently, there are three publications about the genome of Mansonella, which need to be discussed extensively to place the presented results into the context.

6. PLOS authors have the option to publish the peer review history of their article (what does this mean?). If published, this will include your full peer review and any attached files.

Reviewer #1: **Yes: **Luís-Jorge Amaral

Reviewer #2: No

---

## [Author Response · Author response to Decision Letter 0]

18 Mar 2024

POINT BY POINT RESPONSE TO COMMENTS

EDITORIAL COMMENTS

RESPONSE: The ethics statement has been moved to the methods section (134-142). 

7. We note that Figure 1 in your submission contain [map/satellite] images which may be copyrighted. All PLOS content is published under the Creative Commons Attribution License (CC BY 4.0), which means that the manuscript, images, and Supporting Information files will be freely available online, and any third party is permitted to access, download, copy, distribute, and use these materials in any way, even commercially, with proper attribution. For these reasons, we cannot publish previously copyrighted maps or satellite images created using proprietary data, such as Google software (Google Maps, Street View, and Earth). For more information, see our copyright guidelines: http://journals.plos.org/plosone/s/licenses-and-copyright.

RESPONSE: The map has been revised and does not include any satellite or copyrighted images (Fig 1)

Reviewer #1: 

Introduction:

1. Your introduction comprehensively covers the species, distribution, transmission, clinical manifestations, and the main diagnostic methods of 'Mansonella perstans'. However, it could benefit from a clearer exposition on the current understanding (or lack thereof) of effective treatments for 'M. perstans', drawing a comparison with treatments available for onchocerciasis or lymphatic filariasis (LF) where at least one effective treatment is avaliable.

RESPONSE: The treatment for M. perstans in comparison to the other filarial parasites has now been described in lines 80-84. 

2. The third paragraph, while informative, could be restructured for better flow and clarity.

RESPONSE: The this has been addressed

3. Is there any historical data on 'Mansonella' in Ghana before? Understanding if there was a difference in the epidemiology of the parasite before and after (now) interventions for onchocerciasis and LF (as well as population growth and associated consequences) could be insightful

RESPONSE: Unfortunately, there haven’t been any studies to evaluate the impact of onchocerciasis and LF treatments on Mansonella in Ghana, hence we are unable to provide more information on this.

Methodology:

4. (2.1 Study Site) Figure A could be more well defined, although it is readable. I could only find the Hohoe and Adaklu districts, but not East Akyium district, understandably because the latter was negative for M. perstans; however, Figure 1 should not mention that M. perstans is only present in the two former districts, as it is part of the results, not the methodology.

RESPONSE: This has been corrected. The map has also been revised to include the other districts that were included in the bigger study.

5. (2.3 Sample Processing and Analysis) Please provide clarity on which filarial parasites were screened using Nested PCR (O. volvulus, M. perstans and M. perstans?) and the CO1 gene. Also, are these results presented, even if negative, in the result section?

RESPONSE: The Nested PCR assay was used, and this amplifies all the filarial parasites. However, only samples that were positive for M. perstans microscopy were screened. Notwithstanding, if all the samples were tested by PCR, there could have been more positives. Due to limited resources PCR wasn’t done on all the samples. This has been included as a limitation in the discussion section in lines All analysis were based on samples that tested positive for the ITS-1 gene. CO1 gene has been removed from the manuscript because, all the samples submitted were poorly sequenced. Lines 30, 48 and 178

6. (2.6 Sequence Analysis) The rationale for using the 'Loa Loa' sequence instead of a 'Mansonella perstans' sequence would benefit from explanation. If the choice was due to the lack of high-quality reference sequences from 'M. perstans', this should be explicitly stated.

RESPONSE: Loa loa was used as an outgroup due to low quality references and to allow for ingroup comparison. The write up has been updated to include this reason in lines 282 - 284

Results:

7. The range of microfilariae counts is indeed remarkable and adds significant value to the study. It would be beneficial to include the number of individuals screened for 'M. perstans' in East Akyim District, even if the results were negative.

1074 participants were tested in the East Akyim district and this has been included in the Methods section on line 241.

8. Additionally, the placement of each figure legend immediately following its corresponding description would aid in readability and comprehension instead of all at the end of the Results section.

RESPONSE: The figures have now been repositioned to appear after the just below their description.

Discussion:

9. The first paragraph of the Discussion section might be more appropriate in the Introduction, considering its content.

RESPONSE: The first paragraph of the discussion has now been removed and incorporated in the Introduction.

10. The phylogenetic analysis suggesting no variation between the non-contiguous Hohoe and Adalku districts raises intriguing questions about the potential prevalence of 'M. perstans' in surrounding districts. This could be an interesting avenue for future research to understand the prevalence of the parasite in the Volta region. Moreover, the discussion could be expanded to explore the possibilities of integrating 'M. perstans' surveillance with existing lymphatic filariasis and onchocerciasis programmes in Ghana. It would also be beneficial to discuss the broader public health implications of M. perstans, especially in relation to co-infections with other filarial diseases.

RESPONSE: The discussion has been updated to include the impact of the lack of variations in sequences on transmission in the surrounding districts as well as co-infections with other filarial parasites and the opportunities for integrating M. perstans studies in existing LF/onchocerciasis surveillance programs in the districts (lines 349-354).

Reviewer #2: 

Opoku and Souza analysed M. perstans prevalence in the Volta region of Ghana and revealed low (up to 2.51%) infection rates. Moreover, they sequenced the samples and showed that the samples cluster with the reference sequences from other African countries and Brazil. It is very important to draw attention to this neglected filarial infection but the presented study lacks novelty and intense discussion (e.g., genome analysis). Moreover, the introduction and discussion lacks recent publication and is sometimes misleading. In my opinion, the manuscript need to be carefully re-written and updated before

RESPONSE: Most the sections have been updated and rearranged to help with articulation.

Major comments:

1) Line 80-82: The authors highlight that pathogenicity is not known/was so far not assessed, but the presented data sets do also not tackle this issue. Similar to Debrah et al. the presented manuscript shows the prevalence of M. perstans in a specific region of Ghana. In summary, the infection rates are very low compared to the middle belt of Ghana. The authors need to discuss that fact also in regards to the characteristics of the environment that are needed for M. perstans. For example Wanji and colleagues from Cameroon presented over the last years several publications about that.

RESPONSE: The introduction section has been updated to include prevalence from Debrah et al., 2017 and Philips et al in the middle belt (lines 90-95). The discussion has been updated with reference to environmental characteristics and comparison made with the study areas in this study (lines 331-338).

2) A long this line, the introduction need to be updated, such as Mansonella spp. DEUX is not mentioned (line 49-50), the transmitting vector for M. perstans (C. milnei) have been deciphered in Cameroon (line 50-51), and it should be mentioned that M. perstans has no distinct clinical symptoms and is mostly asymptomatic (line 55-57).

RESPONSE: These have been included in lines 315, 66-68 and 103 respectively.

3) Line 64-67: Of course, mansonellosis is still neglected but the are several studies targeting Mansonella and it cannot be stated that it is only assessed during lymphatic filariasis or onchocerciasis surveillance programmes. Again, Wanji and colleagues specifically target Mansonella epidemiology, immunology and treatment efficacy (showing that ivermectin is not effective - should be included in line 68-70).

RESPONSE: This has been addressed and the introduction has been updated indicating ivermectin is not effective in lines 105, 84)

4) In my opinion, it is important to explain the applied methods more in detail. For example, the used PCR need to be shortly explained instead of only revising to a publication. What can be differentiated with this PCR (specificity/what filarial spp can be discriminated?). A long this line, Fig S1 shows a gel blot but there is no further explanation, such as what are the different bands in the sample lines.

RESPONSE: The DNA and PCR methods have been explained in lines 160 – 215.

S1 has been changed to Fig 2 and now included in the results section and explained (lines 246, 251). 

5) In regards to 4), supplement figures/tables need to be embedded, explained and in order within the text.

RESPONSE: This has been addressed accordingly.

6) How were the 30 samples that were sequences selected from the 39 positive individuals.

RESPONSE: Although there were 39 microscopy positive individuals, only samples that were positive by PCR were sequenced. This has been clarified in the text in line 213

7) line 229-230: There are several studies showing the impact of M. perstans on concomitant infections (Muhangi et al. 2007; Hillier et al. 2008; Stensgaard et al. 2016), which can be explained by the strong immune modulation of M. perstans (Ritter et al. 2018).

RESPONSE: This portion has been updated and the appropriate references inserted.

8) Recently, there are three publications about the genome of Mansonella, which need to be discussed extensively to place the presented results into the context.

RESPONSE: Recently published sequences have been mentioned and discussed. 

**The line numbers indicated are in reference to the Revised Clean version document. 

---

## [Decision Letter · Decision Letter 1]

3 Apr 2024

PONE-D-23-37316R1Identification and characterisation of Mansonella perstans in the Volta Region of GhanaPLOS ONE

Dear Dr. Opoku,

Thank you for submitting your manuscript to PLOS ONE. After careful consideration, we feel that it has merit but does not fully meet PLOS ONE’s publication criteria as it currently stands. Therefore, we invite you to submit a revised version of the manuscript that addresses the points raised during the review process.

Although I am satisfied that you have successfully delt with all of the technical concerns raised by the reviewers, there are still a number of minor presentational issues that still need to be addressed for the article to meet the publication criteria of PLOS ONE. Thus while I am not expecting to need to send the manuscript out to reviewers again, I do need you to address the minor remaining issues raised by reviewer 2 and also to address the minor issues I have identified and listed below.

LINES 43-44: Despite its widespread distribution, it is the least studied among all the filarial infections.”LINES 50-53; Change to: “ Prior to sequencing, genomic DNA was extracted and a nested PCR reaction carried out. A total of 30 samples that were positive from the nested PCR with a band size of ~312bp were submitted for Sanger sequencing.LINES 71-71: Please delete this line: Some mosquito species including *Aedes* and *Anopheles* have also been implicated (4, 10, 11). This is miss-leading and is not supported by the cited references.LINE 72-74: “*M. ozzardi* is the major species in Latin America and the Caribbean, while *M. perstans* and *M. streptocerca* are reported in Africa.” This requires a reference, I suggest: Portela CS, Mendes de Araújo CP, Moura Sousa P, Gomes Simão CL, Silva de Oliveira JC, Crainey JL.Filarial disease in the Brazilian Amazon and emerging opportunities for treatment and control.Curr Res Parasitol Vector Borne Dis. 2023 Dec 23;5:100168. doi: 10.1016/j.crpvbd.2023.100168. eCollection 2024.LINE 88: *Wolbachia* is a genus name and thus should always be in italics.LINE 94: “In Ghana, Mansonellosis was first reported in the 1991 by Awadzi and colleagues (25)”. This line needs to be changed. Mansonellosis infections are not only caused by M. ozzardi and M. perstans they are also caused by M. streptocerca and the first reports of M. streptocerca were made from samples taken in Ghana (see Macfie and Corson 1922).Macfie, J. W. S., & Corson, J. F. (1922). A new species of filarial larva found in the skin of natives in the Gold Coast. Annals of Tropical Medicine & Parasitology, 16(4), 465-471.LINES 133-114: ) “as well as vaccine uptake (1, 5, 36).” Should it not be efficacy instead of uptake?LINES 152-153:Change to: ”A, showing the map of Ghana with the districts where samples were collected.”LINES 155-156: This does not make sense, please revise: “shows the Hohoe district (Volta Region) in orange with communities in grey and yellow circles for M. perstans negative and positives respectively that were positive for”. Please also always put *M. perstans* in italics.LINES 174-176: Change to: “The liquid containing DNA was transferred (whilst ensuring that no particles of the filter paper is pipetted with the sample) to a Zymo-Spin™ IICR Column placed in a Collection Tube.”LINES 212-213: “visualised under blue led light.” Should this be UV light?LINE 260: Change to: Fig 2. Gel electrophoresis photograph showing Nest II PCR products.

We look forward to receiving your revised manuscript.

Kind regards,

James Lee Crainey, Ph.D.

Academic Editor

PLOS ONE

Journal Requirements:

Reviewers' comments:

Reviewer's Responses to Questions

**Comments to the Author**

1. If the authors have adequately addressed your comments raised in a previous round of review and you feel that this manuscript is now acceptable for publication, you may indicate that here to bypass the “Comments to the Author” section, enter your conflict of interest statement in the “Confidential to Editor” section, and submit your "Accept" recommendation.

Reviewer #1: All comments have been addressed

Reviewer #2: All comments have been addressed

2. Is the manuscript technically sound, and do the data support the conclusions?

Reviewer #1: Yes

Reviewer #2: Yes

3. Has the statistical analysis been performed appropriately and rigorously? 

Reviewer #1: Yes

Reviewer #2: N/A

4. Have the authors made all data underlying the findings in their manuscript fully available?

Reviewer #1: Yes

Reviewer #2: Yes

5. Is the manuscript presented in an intelligible fashion and written in standard English?

Reviewer #1: Yes

Reviewer #2: Yes

6. Review Comments to the Author

Reviewer #1: The manuscript has undergone significant improvements since the first round of peer review. Notably, the introduction now offers a comprehensive overview, effectively delineating the parasite's life cycle, geographical distribution, and clinical manifestations. Furthermore, the link between mansonellosis overall and its specific relevance to Ghana has been elucidated, providing valuable context for the study. In terms of content, the methods section has been substantially expanded. Additionally, the discussion has been refined and strengthened. Overall, the manuscript has evolved positively, addressing the main concerns raised in the previous review rounds. Below are some minor points, based on the draft with tracked changes.

Abstract:

Change: Lao loa -> Loa loa (line 24)

Suggestion: Occurrence -> Co-occurrence (line 24)

Remove: "Hence," (line 25)

Change: distribution or mansonellosis -> distribution of mansonellosis (line 26)

Change: giemsa -> Giemsa (Capitalize the "G", line 31)

Clarify: lines 37-38 change to "...supporting the hypothesis that M. perstans was introduced into the Americas during the transatlantic slave trade from sub-Saharan Africa"

Remove: Capitalization of "TransAtlantic"

Introduction:

Change: some part -> parts (line 63)

Rephrase: lines 67 and 73, Consider combining sentences or using a different structure to avoid repetition about M. perstans being in Africa.

Define: line 84, MDA (mass drug administration) on first use.

Italicize: line 88, Wolbachia

Remove: Comma after "age" in line 103

Lowercase: "Mansonellosis" unless it starts a sentence.

Methods & Results:

Clarify: Line 202, "Collared" might be unclear. Consider changing to "collected" for better understanding.

Discussion Point: Lines 316-318 may be more appropriate for the discussion?

Discussion:

Rephrase: Lines 396-402 can be rephrased to avoid repetition.

Change: line 409, suggest clarifying "supports work" -> "supports findings"

Italicize: M. perstans (line 420)

Change: Vola -> Volta (line 431)

Additional Point: Briefly mention the possibility of low prevalence due to past ivermectin distribution programs in Ghana. (paragraph starting at line 427).

Clarify: 6. I agree with the authors there should be a more comprehensive guide on the management and surveillance of M. perstans infections. However, in lines 443-450, it is not clear how low levels of M. perstans are alarming (as the prevalence found was low, albeit the limited sample size). Only if assumed that the ivermectin delivery in the past had a substantial impact on the prevalence of the parasite and, with onchocerciasis and LF elimination, ivermectin interruption could lead to the reestablishment of more M. perstans infections. Moreover, the link with malaria is not clear, despite the possible immunomodulation (as other filariae), which could make the host more susceptible to infectious diseases overall (potentially including malaria).

Overall, a brief revision for punctuation would benefit the study.

Reviewer #2: I thank the authors for the responses and clarification. In my opinion the manuscript have been improved and thus I endorse publication.

7. PLOS authors have the option to publish the peer review history of their article (what does this mean?). If published, this will include your full peer review and any attached files.

Reviewer #1: **Yes: **Luís-Jorge Amaral

Reviewer #2: No

---

## [Author Response · Author response to Decision Letter 1]

6 May 2024

1. LINES 43-44: Despite its widespread distribution, it is the least studied among all the filarial infections.”

Response: This has been corrected on line 44.

2. LINES 50-53; Change to: “ Prior to sequencing, genomic DNA was extracted and a nested PCR reaction carried out. A total of 30 samples that were positive from the nested PCR with a band size of ~312bp were submitted for Sanger sequencing.

Response: Sequences has been changed to sequencing on line 54.

3. LINES 71-71: Please delete this line: Some mosquito species including Aedes and Anopheles have also been implicated (4, 10, 11). This is miss-leading and is not supported by the cited references.

Response: This line has been deleted.

4. LINE 72-74: “M. ozzardi is the major species in Latin America and the Caribbean, while M. perstans and M. streptocerca are reported in Africa.” This requires a reference, I suggest: Portela CS, Mendes de Araújo CP, Moura Sousa P, Gomes Simão CL, Silva de Oliveira JC, Crainey JL.Filarial disease in the Brazilian Amazon and emerging opportunities for treatment and control.Curr Res Parasitol Vector Borne Dis. 2023 Dec 23;5:100168. doi: 10.1016/j.crpvbd.2023.100168. eCollection 2024.

Response: The sentence has been updated to “Of the three species, M. ozzardi is the major species in Latin America and the Caribbean, while M. perstans and M. streptocerca are reported in Africa” and the suggested reference included on lines 69-70.

5. LINE 88: Wolbachia is a genus name and thus should always be in italics.

Response: Wolbachia has been italicised on line 89.

6. LINE 94: “In Ghana, Mansonellosis was first reported in the 1991 by Awadzi and colleagues (25)”. This line needs to be changed. Mansonellosis infections are not only caused by M. ozzardi and M. perstans they are also caused by M. streptocerca and the first reports of M. streptocerca were made from samples taken in Ghana (see Macfie and Corson 1922). Macfie, J. W. S., & Corson, J. F. (1922). A new species of filarial larva found in the skin of natives in the Gold Coast. Annals of Tropical Medicine & Parasitology, 16(4), 465-471.

Response: The statement has been corrected and now reads “The first report of mansonellosis was in 1922 where, M. streptocerca was identified in the skin of indigenes from Gold Coast, now Ghana” and the reference included on lines 95-96.

7. LINES 133-114: ) “as well as vaccine uptake (1, 5, 36).” Should it not be efficacy instead of uptake?

Response: “uptake” has been replaced with “efficacy”

9. LINES 152-153:Change to: ”A, showing the map of Ghana with the districts where samples were collected.” 

10. LINES 155-156: This does not make sense, please revise: “shows the Hohoe district (Volta Region) in orange with communities in grey and yellow circles for M. perstans negative and positives respectively that were positive for”. Please also always put M. perstans in italics.

Response: Lines have been changed and reads “A, showing the map of Ghana with the districts where samples were collected. B, shows the East Akyim district (comprising the Abuakwa North and South Districts (Eastern Region)) in green with communities sampled in pale orange circles. C, shows the Hohoe district (Volta Region) in orange. D, shows three districts (Adaklu, Agotime Ziope and Central Tongu Districts in purple, grey and brown respectively) also in the Volta Region. Communities sampled are in circles with yellow and grey indicating sites with M. perstans positives and negatives respectively” on lines 157 -163.

M. perstans is now italicised on line 163.

10. LINES 174-176: Change to: “The liquid containing DNA was transferred (whilst ensuring that no particles of the filter paper is pipetted with the sample) to a Zymo-Spin™ IICR Column placed in a Collection Tube.”

Response: the sentence has been modified and now reads “The liquid containing the DNA was transferred to a Zymo-Spin™ IICR Column placed in a Collection Tube (ensure that no particles of the filter paper is pipetted with the sample).” Lines 178-180.

11. LINES 212-213: “visualised under blue led light.” Should this be UV light?

Response: The gel was visualised with a blue light transilluminator not the usual UV light using the Dual LED Blue/White Light Transilluminator machine from Bio-Helix. The name of the machine has been indicated on line 216.

12. LINE 260: Change to: Fig 2. Gel electrophoresis photograph showing Nest II PCR products.

Response: Gel electrogram has been changed to Gel electrophoresis photograph on line 263.

Reviewer #2

Abstract:

1. Change: Lao loa -> Loa loa (line 24)

Response: Lao has been changed to Loa on line 24

2. Suggestion: Occurrence -> Co-occurrence (line 24)

Response: Occurrence has been changed to co-occurrence on line 23

3. Remove: "Hence," (line 25)

Response: This has been removed. Also, the sentence has been modified to now read “There are few surveillance programmes for assessing the distribution of mansonellosis, due to the associated mild to no symptoms experienced by infected people” on lines 24-25.

4. Change: distribution or mansonellosis -> distribution of mansonellosis (line 26)

Response: “or” has been changed to “of” on line 25.

5. Change: giemsa -> Giemsa (Capitalize the "G", line 31)

Response: giemsa has been correct to start with a capital “G” on line 30.

6. Clarify: lines 37-38 change to "...supporting the hypothesis that M. perstans was introduced into the Americas during the transatlantic slave trade from sub-Saharan Africa"

Remove: Capitalization of "TransAtlantic"

Response: Since there is no strong evidence in this study to draw this conclusion, the sentences have been modified to “Phylogenetic analysis of 194 nucleotide positions showed no differences in the samples collected. The similarities suggests that there could be one species in this area. However, more robust studies with larger sample sizes are required to draw such conclusions. We also observed a clustering of the samples from Ghana with reference sequences from Africa and Brazil, suggesting they could be related. This study draws further attention to a neglected infection, presents the first characterisation of M. perstans in Ghana and calls for more population-based studies across different geographical zones to ascertain species variations and disease distribution”. Lines 34-40.

Introduction:

7. Change: some part -> parts (line 63)

Response: “some part” has been changed to “parts”. 

8. Rephrase: lines 67 and 73, Consider combining sentences or using a different structure to avoid repetition about M. perstans being in Africa.

Response: The paragraph has been modified accordingly.

9. Define: line 84, MDA (mass drug administration) on first use.

Response: The MDA has been written in full on line 85.

10. Italicize: line 88, Wolbachia

Response: This has been done on line 89.

11. Remove: Comma after "age" in line 103

Response: the comma has been removed on line 106.

12. Lowercase: "Mansonellosis" unless it starts a sentence.

Response: This has been corrected on line 111 and 114.

Methods & Results:

13. Clarify: Line 202, "Collared" might be unclear. Consider changing to "collected" for better understanding.

Response: This has been corrected on line 158

14. Discussion Point: Lines 316-318 may be more appropriate for the discussion?

Response: This has been addressed in the discussion section.

Discussion:

15. Rephrase: Lines 396-402 can be rephrased to avoid repetition.

Response: These lines have been reviewed

16. Change: line 409, suggest clarifying "supports work" -> "supports findings"

Response: This section has been reviewed and now reads “The clustering of the Ghana samples and other sequences from Gabon and Sierra Leone with the M. perstans ITS-1 (LT623911.1) from Brazil Amazona State (Fig 4) and other countries in Africa provides evidence of association. supports Findings work by of Tavares da Silva and colleagues (25) that also showed that a recently published genome of M. perstans from Brazil shared similarity with those from Africa.”

Italicize: M. perstans (line 420)

17. Change: Vola -> Volta (line 431)

Response: This has been corrected on line 355.

18. Additional Point: Briefly mention the possibility of low prevalence due to past ivermectin distribution programs in Ghana. (paragraph starting at line 427).

Response: Possibility of IVM contributing to low prevalence and the consequences of stopping IVM treatment in these areas has been incorporated on lines 344-346.

19. Clarify: 6. I agree with the authors there should be a more comprehensive guide on the management and surveillance of M. perstans infections. However, in lines 443-450, it is not clear how low levels of M. perstans are alarming (as the prevalence found was low, albeit the limited sample size). Only if assumed that the ivermectin delivery in the past had a substantial impact on the prevalence of the parasite and, with onchocerciasis and LF elimination, ivermectin interruption could lead to the reestablishment of more M. perstans infections. Moreover, the link with malaria is not clear, despite the possible immunomodulation (as other filariae), which could make the host more susceptible to infectious diseases overall (potentially including malaria).

Response: This portion has been reviewed and emphasis is now on the possibility of resurgence should IVM be discontinued in this area. Lines 343-347. Also, the impact on malaria infections have been clarified on lines 360-373.

20. Overall, a brief revision for punctuation would benefit the study.

Response: The manuscript has been reviewed and punctuations checked. 

Editorial Comments

Response: The figures were uploaded to PACE before uploading them to the submission portal.

Response: The laboratory protocols have now been deposited to protocls.io and available at http://dx.doi.org/10.17504/protocols.io.x54v928b4l3e/v1. This has been referred to in the method section on lines 213 and 226.

Journal Requirements:

Response: The references have been checked for completeness and is no retracted publication.

4 new references have been included.

a. Portela CS, Mendes de Araújo CP, Moura Sousa P, Gomes Simão CL, Silva de Oliveira JC, Crainey JL.Filarial disease in the Brazilian Amazon and emerging opportunities for treatment and control.Curr Res Parasitol Vector Borne Dis. 2023 Dec 23;5:100168. doi: 10.1016/j.crpvbd.2023.100168. eCollection 2024.

b. Macfie, J. W. S., & Corson, J. F. (1922). A new species of filarial larva found in the skin of natives in the Gold Coast. Annals of Tropical Medicine & Parasitology, 16(4), 465-471.

c. Agbemafle E, Kubio C, Ameme DK, Kenu E, Sackey S, Afari E. Evaluation of malaria surveillance system, Adaklu District, Volta Region, Ghana, 2019. International Journal of Infectious Diseases. 2020;101:412.

d. Ejigu BA, Wencheko E. Spatial Prevalence and Determinants of Malaria among under-five Children in Ghana. 2021.

The raw uncropped gel picture has been uploaded.

The keys words list has been updated and now include, Phylogenetic tree and Volta Region

---

## [Editor Report · Decision Letter 2]

10 May 2024

Identification and characterisation of Mansonella perstans in the Volta Region of Ghana

PONE-D-23-37316R2

Dear Dr. Opoku,

We’re pleased to inform you that your manuscript has been judged scientifically suitable for publication and will be formally accepted for publication once it meets all outstanding technical requirements.

Kind regards,

James Lee Crainey, Ph.D.

Academic Editor

PLOS ONE

Additional Editor Comments (optional):

Congratulations on your fine contribution to the field! I look forward to reading more of your work in the future!

---

## [Editor Report · Acceptance letter]

29 May 2024

PONE-D-23-37316R2 

PLOS ONE

Dear Dr. Opoku, 

I'm pleased to inform you that your manuscript has been deemed suitable for publication in PLOS ONE. Congratulations! Your manuscript is now being handed over to our production team.

Kind regards, 

on behalf of

Dr. James Lee Crainey 

Academic Editor

PLOS ONE